# Research Progress on the Protective Effect of Brown Algae-Derived Polysaccharides on Metabolic Diseases and Intestinal Barrier Injury

**DOI:** 10.3390/ijms231810784

**Published:** 2022-09-15

**Authors:** Ying Yang, Meina Liang, Dan Ouyang, Haibin Tong, Mingjiang Wu, Laijin Su

**Affiliations:** 1College of Life and Environmental Science, Wenzhou University, Wenzhou 325035, China; 2Zhejiang Provincial Key Laboratory for Water Environment and Marine Biological Resources Protection, Wenzhou University, Wenzhou 325035, China

**Keywords:** brown algae-derived polysaccharides, main structure specifications, intestinal health, mechanism of action

## Abstract

In the human body, the intestine is the largest digestive and immune organ, where nutrients are digested and absorbed, and this organ plays a key role in host immunity. In recent years, intestinal health issues have gained attention and many studies have shown that oxidative stress, inflammation, intestinal barrier damage, and an imbalance of intestinal microbiota may cause a range of intestinal diseases, as well as other problems. Brown algae polysaccharides, mainly including alginate, fucoidan, and laminaran, are food-derived natural products that have received wide attention from scholars owing to their good biological activity and low toxic side effects. It has been found that brown algae polysaccharides can repair intestinal physical, chemical, immune and biological barrier damage. Principally, this review describes the protective effects and mechanisms of brown algae-derived polysaccharides on intestinal health, as indicated by the ability of polysaccharides to maintain intestinal barrier integrity, inhibit lipid peroxidation-associated damage, and suppress inflammatory cytokines. Furthermore, our review aims to provide new ideas on the prevention and treatment of intestinal diseases and act as a reference for the development of fucoidan as a functional product for intestinal protection.

## 1. Introduction

The intestine is the largest digestive and immune organ in the body, where nutrients are digested and absorbed. Immune cells in the gut can interact with microorganisms to maintain homeostasis in the intestinal environment, in addition to monitoring, recognizing, and differentiating food antigens from pathogens [1,2]. In recent years, intestinal health issues have gained attention and many studies have shown that oxidative stress, inflammation, malnutrition, medications, intestinal barrier damage, and imbalance of intestinal microbiota may cause a range of intestinal diseases, amid other disease problems. The intestinal barrier comprises physical, chemical, immune, and microbial barriers, which work to prevent pathogens and endotoxins from crossing into other tissues, organs, and blood. However, when the intestinal barrier is damaged via intestinal barrier injury (IBI), enterogenic infections, multi-organ failure, inflammatory bowel disease (IBD), sepsis, etc., may occur [3,4]. The intestinal tract is colonized by tens of trillions of microorganisms, referred to as intestinal microbiota, which are involved in digestion and the regulation of immunity, and act as biological barriers to protect the gastrointestinal tract from pathogens. There have been many studies which have shown that an imbalance of intestinal microbiota is associated with various immune-based, gastrointestinal, and metabolic diseases [5,6,7,8,9]. Therefore, maintaining the balance and health of intestinal microbiota, and protecting the integrity of the intestinal barrier are of great importance for the prevention and treatment of various diseases.

Seaweeds are the most abundant resources in the ocean; they can be classified into three main groups based on their pigmentation and chemical composition: brown, red, and green algae [10]. Brown algae are photosynthetic multicellular organisms that include the genera *Laminaria* (*Kombu)*, *Macrocystis*, *Kelp* [11]. Brown algae-derived polysaccharides are one of the main active components of brown algae and mainly include alginate, fucoidan, and laminaran [12,13]. The polysaccharides have been reported to elicit protective effects on the intestinal tract by regulating intestinal microbiota, upregulating the expression of TJ proteins, inhibiting the expression of inflammatory factors, and suppressing oxidative stress, to repair intestinal barrier injury [14,15,16,17]. This paper reviews the protective effects and mechanisms of brown algae-derived polysaccharides in the intestine.

## 2. Brown Algae Polysaccharides

In addition to their food value, other important components found in brown algae include phenolic compounds, sulfated polysaccharides, quinones and several secondary metabolites, all of which have been studied for use against a variety of diseases [18]. Brown algae polysaccharides are collectively referred to as polysaccharides isolated and extracted from brown algae mainly include alginate, fucoidan, and laminaran. The main species and distribution of brown algae with protective effects on the intestine are shown in Table 1.

### 2.1. Classification of Brown Algae Polysaccharides

#### 2.1.1. Alginate

Alginate, with the molecular formula C_6_H_7_NaO_6_, is the main polysaccharide component of both the cell wall and intercellular matrix of brown algae. It is a linear polysaccharide consisting of two conformational isomeric residues, β-D-mannuronate (M) and α-L-guluronate (G), connected by 1,4-glycosidic bonds [36]. The molecule can be arranged in three different conformations: MM, GG, and MG [37]. The structure of alginate is shown in Figure 1 [38,39]. The M/G ratio greatly affects the physicochemical properties, applications, and activities of alginate [40]; molecules with a high G content form firm gels for the food and cosmetic industries, while a high M content results in low viscosity appropriate for the production of nanoparticles, paper, for dyeing, or in the textile industry [37]. Alginate is readily fermented by intestinal bacteria to produce large amounts of short-chain fatty acids (SCFAs) to provide energy to intestinal epithelial and immune cells [41] and affects glucose and lipid metabolism, alleviating obesity, diabetes, insulinemia and other metabolic diseases [42]. Therefore, alginate has an important role in maintaining intestinal health and preventing the development of metabolic diseases. In addition, derivatives of alginate have a wide range of biological activities, and sulfated polymannuroguluronate (SPMG) is a sulfated form of sodium alginate (Figure 2) [43]. SPMG was shown to improve chemotherapy-induced leukopenia [44] and also effectively prevented the internalization of viral particles by interfering with the interaction between viral and host cell receptors [45,46]; SPMG significantly increases the abundance of *Bifidobacterium* and *Lactobacillus* in the intestine by altering the intestinal microbiota structure, prevents diet-induced obesity, and improves glucose tolerance and alleviates inflammation [47].

#### 2.1.2. Fucoidan

Fucoidan, with the molecular formula (C_6_H_10_O_7_S)n, is a polysaccharide containing sulfate groups also known as sulfated polysaccharide. It is mainly present in the cell wall of brown algae where it maintains the stability of the cell membrane and protects its structure from dehydration [48]. There are remarkable differences in the structure and chemical compositions of the fucoidans found in different algal species, but they usually include (1→3)-linked α-L-amylopyranosyl skeletal structures and occasionally both (1→3)-linked and (1→4)-linked α-L-amylopyranosyl structures, where L-amylopyranose residues on C-2 or C-4 can be replaced by sulfate (SO_3_^−^) [49]. Fucoidan has a diverse chemical structure and composition, is usually sulfated and acetylated, and also contains glyoxalate [50,51,52], the structure of which is shown in Figure 3.

Fucoidan has a range of biological activities due to the presence of sulfate groups. Furthermore, it has been demonstrated that fucoidan can improve damage of the intestinal physical, chemical, immune, and biological barriers by upregulating intestinal epithelial cell TJ protein expression, inhibiting intestinal pro-inflammatory cytokine expression, and modulating intestinal microbiota abundance [15,53,54]. 

**Figure 3 ijms-23-10784-f003:**
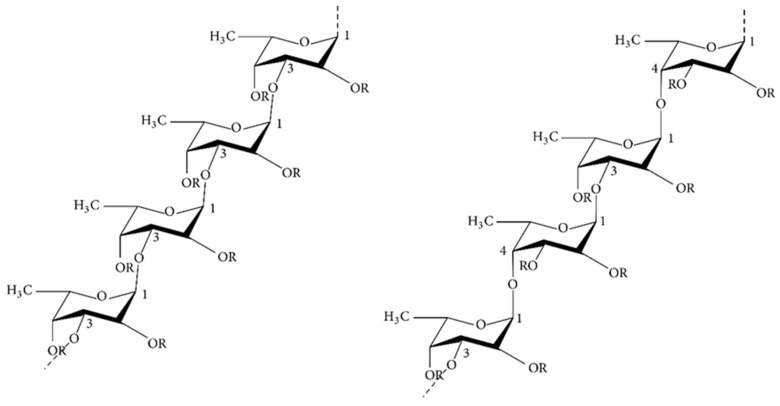
Chemical structures of two different backbones for fucoidan [55].

#### 2.1.3. Laminaran

Laminaran, with the molecular formula C_18_H_32_O_16_ and a molecular weight in the range of 1–10 KDa, is mainly found in the cytoplasm. The main chain consists of β-1,3-D-glucose and β-1,6 as a branched chain [56]. There are two types of chains: one linked by D-mannitol residue ends (M), and the other linked by D-glucose residue ends (G) (Figure 4) [57]. Laminaran has both water-soluble and water-insoluble forms; one form is characterized by complete solubility in cold water, while the other is soluble only in hot water. The presence of branched chains also affects solubility; the more branched chains, the higher the solubility in cold water [58]. Studies have shown that laminaran administration to rats affects the composition of intestinal mucus, increases the production of SCFAs, regulates intestinal metabolism, reshapes the structure of intestinal microbiota and reduces the incidence of obesity, NAFLD and diabetes [59,60].

### 2.2. Extraction of Brown Algae Polysaccharides

Brown algae polysaccharide fractions are complex, and the extraction methods and conditions may affect the composition, yield, molecular weight and biological activity of polysaccharides [62]. Therefore, if we want to obtain the target components efficiently, we have to choose the corresponding extraction methods. The steps of polysaccharide extraction include: removal, extraction and purification [63]. According to the chemical properties of polysaccharides, different extraction methods can be chosen: water extraction, acid extraction, alkali extraction, auxiliary extraction, etc.; according to the chemical properties of polysaccharides, molecular weight size, different purification methods can be chosen: precipitation, column chromatography, etc. [64,65]. The extraction process of brown algae polysaccharide is shown in Figure 5 [66,67,68].

## 3. Protective Effect of Brown Algae-Derived Polysaccharides on Metabolic Diseases and Intestinal Barrier Injury

Many factors affect intestinal health, as shown in Figure 6; inflammation, drugs, lipid peroxidation, and damage to the intestinal barrier can all cause diseases related to intestinal inflammation. The activation of nuclear factor kappa-B (NF-κB) and mitogen-activated protein kinase (MAPK) pathways can promote the secretion of pro-inflammatory cytokines (TNF-α, NO, iNOS, COX-2, IL-1β, IL-6, etc.) by T cells, B cells and monocytes. Excessive accumulation of reactive oxygen species (ROS) can promote the production of inflammatory cytokines by immune cells, leading to oxidative stress and inflammation. This further activates signaling pathways such as MAPK, PKC, JNK and ERK, leading to the breakdown of the TJ complex between epithelial cells, and thus impairing intestinal epithelial barrier function [69]. Additionally, drug treatment can result in damage to the intestinal mechanical barriers [3,70]. 

The protective effects of brown algae-derived polysaccharides on the intestine are shown in Table 2. These polysaccharides not only increase the synthesis of transmembrane proteins in the intestinal epithelial cells, but also regulate the intestinal microbiota, inhibit inflammatory responses, and play a protective and regulatory role against damage to the intestinal barrier and inflammation.

### 3.1. Brown Algae-Derived Polysaccharides Maintain Intestinal Barrier Integrity

The intestinal barriers include physical, chemical, immune, and biological elements; defects in barrier function may lead to chronic immune activation and so play a pathogenic role in many diseases such as celiac disease, colorectal cancer, inflammatory bowel disease (IBD), obesity, and diabetes [81]. Therefore, maintaining intestinal barrier integrity plays an important protective role in intestinal health and disease prevention.

#### 3.1.1. Maintaining the Integrity of the Physical Barrier

The intestinal physical barrier consists of intestinal epithelial cells (IECs) and intercellular junctions: adherens junctions (AJs), bridging granules, and TJs [82]. The TJ is the most apical linker complex, consisting of transmembrane proteins, such as occludin, claudins, and junctional adhesion molecule (JAM), and intracellular plaque proteins, such as zonula occludens (ZO) [83]. These proteins are responsible for closing cellular gaps and regulating selective paracellular ion solute transport. TJs are regulated by several intracellular signaling pathways, such as myosin light chain kinase (MLCK), PKC, and MAPK [84]. In addition, the TJ is connected to the cytoskeleton supporting the epithelial cells; thus, forming a dynamic barrier system composed of IECs [85]. In fact, the TJ is the most important intercellular structure, and its disruption and increased paracellular permeability play a crucial role in the pathogenesis of IBD [3]. Furthermore, the length of small intestinal villi and crypt depth are important indicators of small intestinal digestion and absorption. The longer the length of the villi and the shallower the depth of the crypts, the better the digestion and absorption. Moreover, studies have shown that fucoidan can improve constipation, improve small intestinal tissue morphology, and repair tissue damage of IECs by promoting intestinal motility [17,86]. 

Matayoshi et al. [86] used a double-blind randomized clinical trial to study the efficacy of dried powder of *Cladosiphon okamuranu* in improving constipation. The results showed that *C. okamuranu* increase the frequency of defecation and regulate bowel function in patients with constipation. Xue et al. [17] showed that the consumption of fucoidan improved the structure of the small intestinal villi in mammary carcinoma rats, upregulated the levels of TJ proteins, ERK1/2, and p38 MAPK phosphorylation in rat jejunum tissues. It has also been demonstrated that fucoidan significantly reduced paracellular permeability and enhanced intestinal barrier function by upregulating claudin-1, Occludin, and ZO-1 expression [15,87]. Alginate oligosaccharides increased occludin abundance in TNF-α-treated IPEC-J2 cells, thereby ameliorating inflammatory damage in intestinal epithelial cells [88]. Thus, brown algae-derived polysaccharides can improve intestinal physical barrier damage by upregulating transmembrane protein expression, activating signaling pathways related to the regulation of TJs, and decreasing paracellular permeability.

#### 3.1.2. Enhancement of Intestinal Chemical Barrier Function

The intestinal chemical barrier includes digestive juices, bile acids, antimicrobial peptides (AMPs), mucins, and other compounds secreted by IECs in the intestinal lumen to prevent bacterial adhesion [89]. Goblet cells can secrete mucins to form a dense gel covering the intestinal mucosa to prevent bacterial invasion [90]. MUC2 is the most abundant mucin in the small intestine and colon, and MUC2 deficiency makes the host susceptible to pathogenic bacteria invasions, alters mucus layers, defective AMPs, and can lead to bacterial translocation due to increased intestinal permeability, resulting in disease [91,92].

The addition of the brown algae-derived polysaccharides promoted intestinal mucin expression, and A.G. Smith found that the addition of laminarin to the diet had a significant effect on the expression of secreted MUC2 and the membrane-bound mucin MUC4 in the pig colon [93]. In addition, polysaccharides can bind bile acids, inhibit bile acid reabsorption, and promote cholesterol metabolism in the liver. Related studies have shown that the ability of polysaccharides to bind bile acids may be related to the sulfate groups in polysaccharides [94].

#### 3.1.3. Improving Intestinal Immune Barrier Protection

The intestine is the largest immune system in the body, and the gut-associated lymphoid tissue (GALT) is the main component of the intestinal immune system, accounting for 70% of the systemic immune function. Intestinal immune cells include T cells, B cells, innate lymphocytes (ILC1, ILC2, ILC3), and the mononuclear phagocyte system (monocytes, dendritic cells (DC), and macrophages) [95,96]. These immune cells can secrete relevant pro- and anti-inflammatory cytokines and enzymes to regulate intestinal inflammation and immune responses [97]. Specific secretory immunoglobulins (sIgA) are mainly produced by lymphocytes and plasma cells, which are distributed on the surfaces of the intestinal mucosa. They are the most abundant immunoglobulins in intestinal secretions, functioning as the main factors stopping pathogen invasion, and so playing a key role in the intestinal immune system. Fucoidan has bi-directional immunomodulatory effects: immune-enhancing and immune-suppressing. The pattern recognition receptors (PRRs) that recognize polysaccharides in the intestine include Toll-like receptors (TLRs) and nucleotide oligomerization domain (NOD)-like receptors (NLRs). The specific recognition of polysaccharides by PRRs may trigger signaling pathways that regulate inflammation and immune cytokines, ultimately leading to upregulation of related gene expression and protein synthesis; thus, regulating the intestinal immunity to protect against compromised intestinal immune barriers [97].

Cancer is one of the most serious threats to human health, and CPA is commonly used to treat cancer, but its long-term use can suppress normal immune responses [98]. CPA significantly increases ROS levels and promotes apoptosis; it reduces the production of antioxidant enzymes (Catalase (CAT), Superoxide Dismutase (SOD), Glutathione peroxidase (GSH-Px)) in the spleen and thymus [99]. The fucoidan derived from *Acaudina molpadioides* can reduce intestinal inflammation, promote TJ protein expression, and increase the abundance of SCFAs-producing microorganisms (*Coprococcus*, *Rikenella*, and *Butyricicoccus*) by reducing the CPA-induced intestinal mucosal damage [54]. Additionally, it has been reported that *Laminaria* sodium alginate increased immune organ indices, decreased splenic T lymphocytes, and significantly increased serum immunoglobulin and cytokine secretion in immunosuppressed mice [73]. Fucoidan reversed CPA-induced immunosuppression, promoted intestinal immunity, and increased sIgA secretion [100]. Therefore, fucoidan can improve the damage to the intestinal immune barrier by promoting the secretion of intestinal sIgA and reducing intestinal inflammation-related cytokines; thus, protecting the intestine.

The inflammatory response is an excessive immune response; both LPS and dextran sodium sulfate (DSS) induce inflammatory responses. Jeong et al., [74] showed that fucoidan significantly inhibited LPS-induced secretion of pro-inflammatory mediators, including NO, PGE2, TNF-α, and IL-1β, without any remarkable cytotoxicity. Fucoidan also inhibited NF-κB translocation from the cytoplasm to the nucleus and attenuated LPS-induced intracellular ROS production in RAW 264.7 macrophage-like cells. The pro-inflammatory cytokine IL-6 is highly expressed in the serum of patients with Crohn’s disease (CD), and blocking IL-6 expression may provide benefit to patients with CD. It has been shown that IL-6 plays an important role in colitis in mice and accordingly, inhibition of IL-6 secretion plays a key role in the treatment of intestinal inflammation [101]. In a mouse model of chronic colitis, treatment with fucoidan (*Cladosiphon okamuranus Tokida*) inhibited the activation of NF-κB pathway, thereby suppressing IL-6 synthesis and down-regulating IFN-γ expression to improve chronic colitis in rats [102]. Fucoidan (20 mg kg^−1^) could inhibit IL-6, TNF expression, and IFN-γ to prevent gastric inflammation associated with aspirin-induced gastric ulcers [103]. It has been reported that low concentrations of fucoidan promote the secretion of NO, iNOS, ROS, IL-1β, IL-6, IL-12, and TNF-α by macrophages, while high concentrations of fucoidan inhibit the secretion of pro-inflammatory factors and thus act as an anti-inflammatory agent [100]. Alginate oligosaccharides inhibited apoptosis and decreased IL-6 and TNF-α concentrations in TNF-α-treated IPEC-J2 cells [88]. Bioactive molecules, such as NO and TNF-α, can indirectly induce cytotoxicity, but within a certain concentration range, pro-inflammatory factors can induce other immune cells to release cytokines and participate in immune response further activating their anti-tumor function. Therefore, further studies are needed to prove whether the promotion or inhibition of pro-inflammatory cytokine secretion in the bidirectional immunomodulatory effect of brown algae-derived polysaccharides can coordinate innate immunity and inflammatory response.

#### 3.1.4. Maintenance of Intestinal Microbial Barrier by the Intestinal Microbiota Balance

The intestinal tract is colonized by tens of trillions of microorganisms—the intestinal microbiota—that are involved in digestion, modulate immunity, and act as a biological barrier to protect the gastrointestinal tract from pathogens. The intestinal microbiota can maintain host health by regulating metabolism, epithelial barrier integrity, and immune system development and function, intestinal microbiota can maintain host health by regulating metabolism, epithelial barrier integrity and immune system development and function, reducing the risk of obesity, hyperlipidemia and diabetes [104,105,106,107]. Polysaccharides typically promote the proliferation of SCFA-producing microorganisms by modulating immune cells to improve intestinal microbiota and the entry of undigested polysaccharides into the colon for fermentation.

Under normal circumstances, the immune system and intestinal microbiota interact to regulate the body’s immunity and metabolism to help maintain the host’s health, but once this mutual balance is disrupted, it will cause the intestinal microbiota and immune system to become dysregulated, making the body more susceptible to pathogenic infections and inducing various diseases [108]. In patients with IBD, the diversity of the intestinal microbiota is reduced, with fewer *Bacteroides* and an increase in *Proteus* (e.g., *E. coli* and *Clostridium*). The decrease in the number and impaired function of both Paneth and goblet cells leads to a decrease in the thickness of the mucus layer, which in turn reduces mucosal integrity, promotes dysbiosis of the intestinal microbiota, and finally leads to impaired physical barrier function of the intestine and thus bacterial translocation [109]. Dysbiosis of the intestinal microbiota increases bacterial translocation, which stimulates the activation of antigen-presenting cells (e.g., DCs and macrophages), which then induces alterations in T-cell subsets (increased Th1, Th2, and Th17 cells as well as decreased Treg cells), leading to pro-inflammatory responses and tissue damage [97]. Brown algae extracted laminaran and fucoidan were added to the diet of weaned pigs and fed to pigs, respectively, to increase the intestinal crypt depth ratio. A remarkable reduction in the abundance of *Enterobacteriaceae* in the intestine, and down-regulation of IL-6, IL-17A, and IL-1β mRNA expression in the colon was seen [110]. *Acaudina molpadioides* derived fucoidan increased the ratio of intestinal villi length to crypt depth and improved the IFN-γ/IL-4 ratio in an intestinal mucosa mouse model, allowing for Th1/Th2 immune homeostasis. Additionally, fucoidan promoted IgA expression to enhance intestinal adaptive immunity [111].

Another beneficial effect of the polysaccharides on intestinal microbiota is that they contribute to the proliferation of SCFAs-producing microorganisms. The anaerobic fermentation of polysaccharides into the intestine promotes anaerobic metabolism of bacteria to produce SCFAs, thereby lowering the pH of the intestine and inhibiting the proliferation of pathogenic gram-negative bacteria in the intestine [112,113]. Fucoidan exhibits prebiotic activity, which promotes the growth of beneficial intestinal bacteria and inhibits the growth of harmful bacteria. Liu et al. [114] showed that fucoidan from *Undaria pinnatifida* can increase the abundance of intestinal *Bacteroidetes* and *Alloprevotella* in BALB/c mice, decrease the abundance of *Firmicutes*, *Staphylococcus,* and *Streptococcus*, lower serum and liver cholesterol levels, and alleviate dyslipidemia in mice on a high-fat diet; fucoidan improved symptoms in diabetic mice associated with altered intestinal microbiota, reducing the relative abundance of diabetes-associated intestinal microbiota such as *Oscillibacter*, *Ruminococcaceae*, *Peptostreptococcaceae*, and *Peptococcaceae* [115]; fucoidan may also improve insulin resistance by reshaping the structure of the intestinal microbiota [25]. Alginate oligosaccharides decreased the relative abundance of bacteria of the phylum *Bacteroidetes* and increased the abundance of phyla *Firmicutes* and *Actinobacteria* in the intestine of mice with ulcerative colitis. These findings suggested that alginate oligosaccharides carried out the function of maintaining the mucosal barrier by modulating the intestinal ecosystem [116].

In addition, intestinal microbiota metabolites (SCFAs) can target the intestine, liver, pancreas and other organs to regulate gastrointestinal hormone secretion, control blood glucose, improve blood lipids, alleviate insulin resistance and inflammation, and have an impact on host physiology and immunity [117,118]. Acetate protects the host intestine from infection, whilst butyrate provides energy to colon cells, regulates stem cell proliferation, anti-inflammatory macrophage polarization with inhibition of histone deacetylase (HDAC), and activation of histone acetyltransferases, thus, affecting cellular transcriptional regulation [108]. SCFAs provide energy to intestinal cells, reduce the incidence of inflammatory diseases, and modulate the innate and adaptive immune systems [119,120]. Studies have shown that in vitro fecal fermentation of sodium alginate from *Ascophyllum nodosum* can significantly increase the concentrations of acetate and propionate [75]. Furthermore, it has been shown that fucoidan can significantly increase the concentrations of acetate and butyrate in the feces of mice with colitis to alleviate antimicrobial-induced colitis in mice [28]. In conclusion, the entry of indigestible fucoidan into the colon for fermentation can increase intestinal microbiota richness, promote the production of SCFAs, and lower intestinal pH to inhibit the proliferation of pathogenic microorganisms; thus, maintaining intestinal microbiota balance and enhancing intestinal immunity.

### 3.2. Inhibition of Lipid Peroxidation Damage by Brown Algae-Derived Polysaccharides

Oxidative stress is one of the factors that disrupt the intestinal barrier. Indeed, oxidative stress due to the accumulation of ROS is the pathological basis of many intestinal diseases. Oxidative stress can disrupt the intercellular TJ complex through multiple signaling pathways, namely, PKC, MAPK, JNK, and ERK; thereby, impairing intestinal epithelial barrier functions by redistributing TJs and AJs [69].

It has been shown that *Fucus vesiculosus* fucoidan (FVF) increased glucose consumption and alleviated sodium palmitate-induced insulin resistance through ROS-mediated JNK and Akt signaling pathways with decreased cellular levels of ROS [31]. *Sargassum fusiforme* alginate can reduce oxidative stress to some extent by increasing the activities of antioxidant enzymes (CAT, SOD) in the serum of high-fat diet-induced diabetic mice [6]. Thus, brown algae-derived polysaccharides may protect the intestine from oxidative stress-induced damage by decreasing ROS production, or by promoting an increase in the activities of antioxidant enzymes.

### 3.3. Inhibition of Inflammatory Cytokines by Brown Algae-Derived Polysaccharides

A moderate immune response protects the intestine from pathogens and eliminates them, whereas an excessive immune response makes the intestine more vulnerable to pathogens. Inflammation is a common excessive immune response in organisms [97]. Endotoxin LPS is capable of eliciting inflammatory responses, and many studies have used LPS to induce inflammatory cells and animal models; LPS is a major ligand for TLR4, and LPS binding to TLR4 can promote the expression of pro-inflammatory cytokines and enzymes, such as TNF-α, NO, iNOS, and COX-2. LPS can also increase pro-inflammatory cytokines and enzymes by activating expression of the MAPK pathway [121]. Therefore, the need to maintain pro-inflammatory and anti-inflammatory cytokines in a certain range is a useful approach for the treatment of inflammatory diseases.

Currently, brown algae-derived polysaccharides have been shown to possess inhibitory activities on the expression of pro-inflammatory cytokines. In LPS-treated RAW 264.7 cells, *Sargassum horneri* fucoidan could downregulate the protein expression levels of iNOS and COX-2, production of TNF-α and IL-1β, and inhibit the phosphorylation of ERK1/2 and p38 in a dose-dependent manner [122]. Fucoidan of *Cladosiphon okamuranus* could inhibit neutrophil extravasation into the peritoneal cavity, eliciting anti-inflammatory effects in a rat model of acute peritonitis [51]. In addition, the fucoidan derived from *Sargassum hemiphyllum* can inhibit LPS-induced inflammatory responses by inhibiting IL-1β and TNF-α, promoting IL-10 and IFN-γ, and significantly enhancing intestinal epithelial barrier and immune function [14]. Therefore, brown algae-derived polysaccharides can inhibit NF-κB and MAPK pathways, and the expression of related pro-inflammatory factors; thus, they have the potential to be developed into functional anti-inflammatory foods.

In summary, brown algae-derived polysaccharides can play both preventive and therapeutic roles in intestinal diseases by repairing intestinal barrier damage, inhibiting lipid peroxidation, and suppressing the expression of inflammatory factors. The protective mechanism of brown algae-derived polysaccharides against intestinal damage is shown in Figure 7. Using brown algae-derived polysaccharides as prebiotics can upregulate TJ protein expression to maintain the integrity of the intestinal physical barrier; thus, reducing bacterial translocation and pathogen invasion. The brown algae-derived polysaccharides regulated intestinal microbiota abundance, increased the abundance of beneficial intestinal bacteria, and promoted the production of SCFAs and AMP along with MUC2 by goblet cells. In addition, they enhanced intestinal chemical barrier function and stimulated DC, T, and B cells to secrete anti-inflammatory cytokines and antibodies, which improved intestinal immunity and strengthened the intestinal microbial barrier. Brown algae-derived polysaccharides can play an immune-enhancing role by activating NF-κB and MAPK signaling pathways, up-regulating inflammatory factor expression to alleviate the immunosuppressive effect induced by the chemotherapeutic drug CPA. The polysaccharides ameliorated production of reactive oxygen species by inhibiting the activation of the NF-κB pathway. This promoted the synthesis of antioxidant enzymes and inhibited the inflammatory response caused by LPS. Simply summarized, the polysaccharides prevented the activation of NF-κB and MAPK signaling pathways and resulted in anti-inflammatory effects through the inhibition of pro-inflammatory cytokines.

## 4. Conclusions and Future Perspectives

The biological activity of brown algae-derived polysaccharides is related to their molecular weight, size, monosaccharide composition, and the functional group content, location, conformation, and configuration [123]. Therefore, the classification of polysaccharide structures and the study of their structure-function relationships can help to reveal the biochemical basis of activity, and to lay the theoretical foundation for the screening of polysaccharides with protective effects on the intestine.

The development of intestinal diseases is often associated with damage to the intestinal barrier, so maintaining the integrity of the barrier plays an important role in protecting the intestine. Among them, the intestinal biological barrier plays a more important role in maintaining intestinal health. Intestinal microbiota, as well as intestinal microbiota metabolites, can affect the integrity of the epithelial barrier, chemical barrier, and immune barrier. Much of the current research on intestinal microbiota has focused on the following: the effects of targeting intestinal microbiota on disease, the effects of drugs and disease on the intestinal microbiota, and the effects of drugs and disease on microbiota diversity, along with the abundance of beneficial and harmful intestinal bacteria. Despite the rich diversity of intestinal microbiota, the effects of only a few microorganisms on disease and immunity have so far been demonstrated, and further studies on the effects of other microorganisms on intestinal disease and immunity are required to understand the relationship between intestinal microbiota and disease.

We have shown that fucoidan exhibits prebiotic activity and can protect and regulate the intestinal barrier. It functions by improving intestinal barrier damage, promoting the proliferation of beneficial intestinal microorganisms, inhibiting the proliferation of intestinal pathogenic bacteria, and promoting the growth of intestinal short-chain fatty acid-producing microorganisms. In summary, fucoidan possesses anti-inflammatory, anti-tumor, and immunomodulatory potentials. It also has the potential to be developed as a functional food for patients with inflammatory bowel disease and cancer.

## Figures and Tables

**Figure 1 ijms-23-10784-f001:**
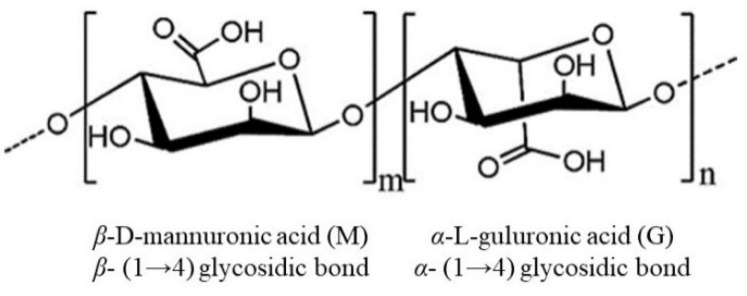
The structure of the alginate.

**Figure 2 ijms-23-10784-f002:**
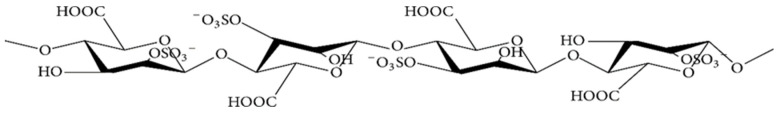
Chemical structure of sulfated polymannuroguluronate (SPMG).

**Figure 4 ijms-23-10784-f004:**
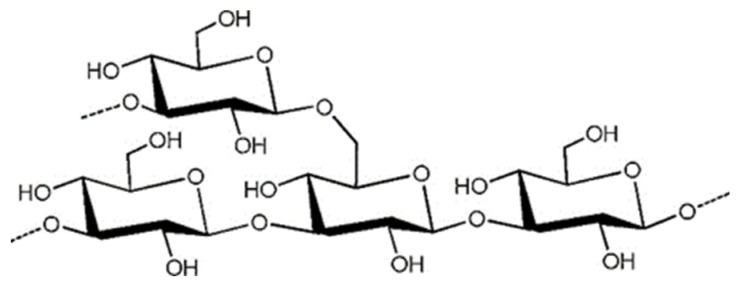
The presumptive structure of the laminaran. (Reprinted with permission from Ref. [61]. Copyright 2020, Elsevier).

**Figure 5 ijms-23-10784-f005:**
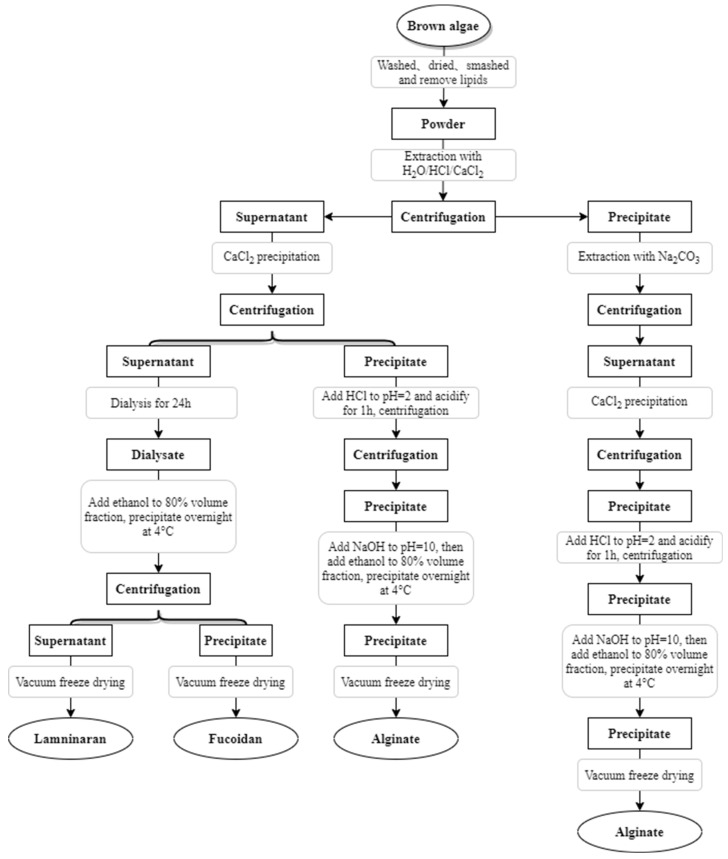
Extraction and separation process of brown algae polysaccharide.

**Figure 6 ijms-23-10784-f006:**
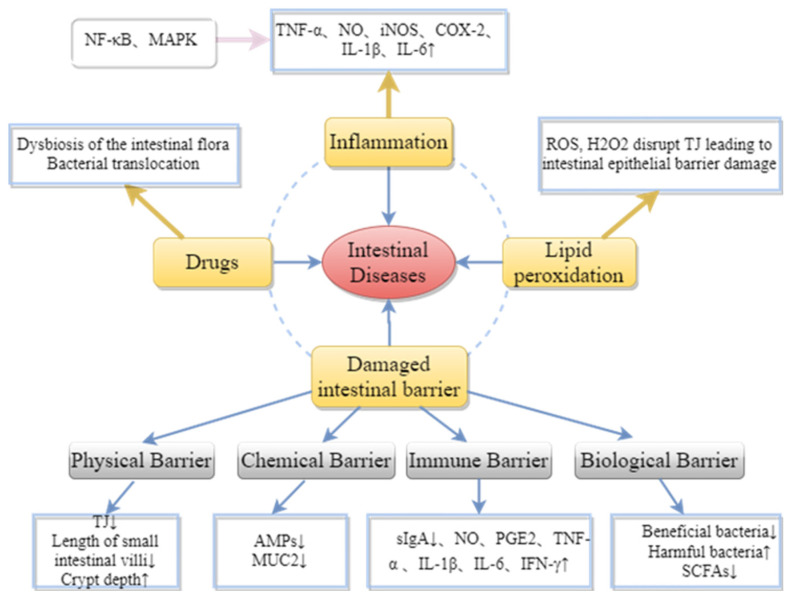
Pathogenesis of intestinal diseases.

**Figure 7 ijms-23-10784-f007:**
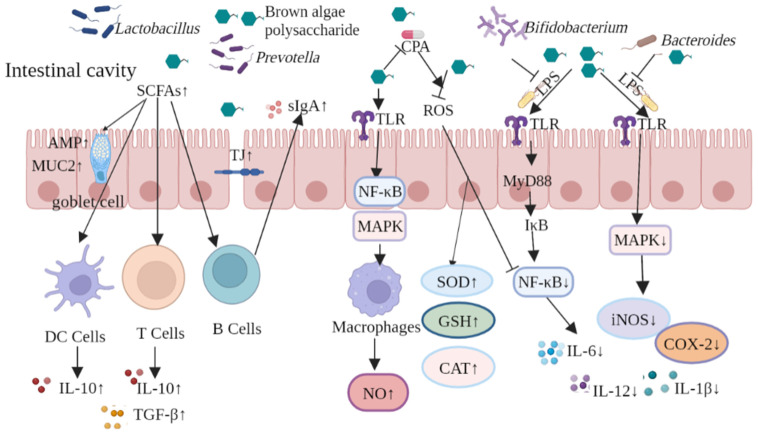
Protective mechanisms of brown algae-derived polysaccharides against intestinal damage.

**Table 1 ijms-23-10784-t001:** The main species and distribution of brown algae with protective effects on the intestine.

Genus	Species	Distribution	References
*Ecklonia*	*Ecklonia radiata, Ecklonia cava*	Distributed in Australia, Korea, China and other countries, in China mainly in Liaodong, Shandong, Zhejiang, Fujian Province.	[19,20,21]
*Sargassum*	*Sargassum fusiforme, Sargassum plagiophyllum, Sargassum thunbergii*	Asian countries are more widely distributed, widely distributed in Fujian Province, Zhejiang Province, China, Korea, Japan.	[22,23,24,25]
*Laminaria*	*Laminaria japonica*	It is mainly distributed in the northwestern Pacific Ocean. Widely distributed in Japan, Russia, China and other countries.	[26,27]
*Ascophyllum*	*Ascophyllum nodosum*	Distributed in the coastal waters of the North Atlantic Ocean, such as Canada, Norway, Ireland, the United Kingdom, France and other countries.	[28,29]
*Fucus*	*Fucus vesiculosus*	Mostly found in tropical and subtropical seas.	[30,31]
*Undaria*	*Undaria pinnatifida*	Mainly in the northwest coast of the North Pacific Ocean, native to China, Japan and the Korean Peninsula.	[32,33]
*Macrocystis*	*Macrocystis pyrifera*	Mainly distributed along the eastern Pacific coast.	[34,35]

**Table 2 ijms-23-10784-t002:** Protective effect and mechanism of brown algae-derived polysaccharides on intestinal health.

Source	Type	Inducer	Models	Function	References
*Cladosiphon okamuranus Tokida*	Fucoidan	H_2_O_2_	Caco-2 cells	Fucoidan remarkably reduced H_2_O_2_-induced paracellular permeability. Up-regulation of endogenous expression of claudin-1, claudin-2, and occludin in Caco-2 cells.	[15]
*Acaudina molpadioides*	Fucoidan	Cyclophosphamide (CPA)	Mice	Fucoidan intervention alleviates inflammation, increases tight junction protein expression, and increases the abundance of *Coprococcus*, *Rikenella*, and *Butyricicoccus.*	[54]
*Laminaria japonica*	Fucoidan	cefoperazone	Mice	Inhibiting the production of pro-inflammatory cytokines, restored the richness and diversity of intestinal microbiota, and improved the structural damage of intestinal mucosa.	[53]
*Cladosiphon okamuranus*	Fucoidan	-	Zebrafish	Down-regulation of the relative expression of the pro-inflammatory gene IL-1β.	[71]
*Ascophyllum nodosum*	Fucoidan	ciprofloxacin-metronidazole	Mice	Increased the abundance of *Ruminococcaceae*, and *Akkermansia*, and decreased the abundance of *Proteus* and *Enterococcus*; inhibited the overproduction of TNF-α, IL-1β, and IL-6, and promoted the expression of IL-10.	[28]
*Sargassum fusiforme*	Sulfated polysaccharide	-	-	*In vitro* fermentation increased the abundance of *Faecalibacterium*, *Phascolarctobacterium*, *Bifidobacterium*, and *Lactobacillus.*	[72]
*Laminaria*	Alginate	CPA	Mice	Up-regulation of tight junction protein expression reduced intestinal mucosal damage, decreased serum D-lactate and lipopolysaccharide concentrations, and downregulated toll-like receptor 4 (TLR4) and mitogen-activated protein kinase (MAPK) pathway expression to reduce intestinal inflammation.	[73]
*Fucus vesiculosus*	Fucoidan	LPS	RAW 264.7 Cell	Inhibited the secretion of NO, PGE 2 and TNF-α, IL-1β and reduced the production of intracellular reactive oxygen species.	[74]
*Ascophyllum nodosum*	Alginate	-	-	Promoted the growth of *Bifidobacteria* and *Lactobacilli*, increased levels of acetate and propionate.	[75]
*Laminaria japonica*	Alginate	-	-	Increased the abundance of *Bacteroides.*	[76]
*Eisenia bicyclis*	Laminaran	-	-	Inhibited ammonia, phenol, and indole production by human fecal microbiota and reduced indole levels in the cecum.	[77]
*Ecklonia radiata*	Polysaccharides	-	-	Increase in total bacteria, *Bifidobacterium*, *Lactobacillus* and increase in total SCFA, acetic and propionic acids.	[19]
*Ecklonia cava*	Fucoidan	LPS	Zebrafish	Inhibiting ROS and NO production induced by LPS treatment to alleviate inflammation.	[20]
*Sargassum fusiforme*	Fucoidan	Streptozotocin	Mice	Fucoidan significantly reduced fasting blood glucose, improved glucose tolerance, reduced oxidative stress in diabetic mice, and increased the abundance of beneficial intestinal microbes including *Bacteroides*, *Faecalibacterium* and *Blautia.*	[24]
*Sargassum fusiforme*	Fucoidan	High-fat diet	Mice	Fucoidan improves HFD-induced insulin resistance by activating the Nrf2 pathway, remodeling the intestinal microbiota, and reducing intestinal inflammation.	[25]
*Laminaria japonica*	Fucoidan	CPA	Mice	Fucoidan increased spleen and thymus indices, increased serum levels of IL-6, IL-1β, TNF-α, and IgG, and improved immunosuppression in mice. Increased the abundance of *Lactobacillaceae* and *Alistipes*, and decreased the abundance of *Erysipelotrichia*, *Turicibacter*, *Romboutsia*, *Peptostreptococcaceae*, and *Faecalibaculum.*	[78]
*Ascophyllum nodosum*	Fucoidan	ciprofloxacin-metronidazole	Mice	Dietary fucoidan prevented colon shortening and alleviated colon tissue damage by increasing the abundance of potentially beneficial bacteria (e.g., *Ruminococcaceae_UCG_014* and *Akkermansia*) and decreasing the abundance of harmful bacteria (e.g., *Aspergillus* and *Enterococcus*), fucoidan also inhibited the overproduction of TNF-α, IL-1β, and IL-6 and promoted the expression of IL-10.	[28]
*Fucus vesiculosus*	Fucoidan	Sodium palmitate	HepG2 Cells Mice	Fucoidan significantly reduced the phosphorylation level of JNK and increased the phosphorylation of protein kinase B (pAkt). It improved hyperglycemia and serum insulin levels in mice with metabolic syndrome.	[31]
*Undaria pinnatifida*	Fucoidan	High-fat diet	Mice	Fucoidan reduces weight gain, fat accumulation and intestinal permeability in mice with metabolic syndrome. Intestinal *Firmicutes* and *Bacteroidetes* in fucoidan-treated high-fat diet mice were restored to normal levels and promoted the production of SCFAS and enhanced the expression level of IL-10.	[32]
*Sargassum thunbergii*	crude polysaccharide	-	-	Increased abundance of intestinal beneficial bacteria such as *Bifidobacterium*, *Roseburia*, *Parasutterella*, and *Fusicatenibacter* by in vitro fecal fermentation after 24 h of fermentation.	[79]
*Laminaria japonica*	Fucoidan	-	Mice	Increased in the abundance of *Ruminococcaceae* and *Lactobacillu* and decreased in the serum levels of lipopolysaccharide-binding protein.	[80]

## Data Availability

The authors confirm that the data supporting the findings of this study are available within the article.

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
