# Peer review of "Research Progress on the Protective Effect of Brown Algae-Derived Polysaccharides on Metabolic Diseases and Intestinal Barrier Injury"

_ijms, 2022, doi:10.3390/ijms231810784_

Round 1

Reviewer 1 Report

Major comment

This review manuscript is well written and useful for readers, especially researchers who will start the study in this field. I could well understand the protective effect of brown algae-derived polysaccharides. If authors revise the manuscript according to following minor comments, I recommend editors to accept this review manuscript to publish on Int J Mol Sci.

Minor comments

 Figure 1

Please show abbreviation “M” and “G” in the Figure as “β-D-mannuronic acid (M)” and “α-L-guluronic acid (G)”.

Line 152

The author describes “The TJ is the most apical linker complex, consisting of four unique transmembrane proteins: occludins, claudins, zonula occludens-1 (ZO-1), and junctional adhesion molecule (JAM)”, but ZO-1 is not transmembrane protein, and it exists in cytoplasm. Please correct that.

 Line 212

Is Cy abbreviation for cyclophosphamide? Should describe non abbreviated word at the first appearance. Moreover, if Cy is abbreviation for cyclophosphamide, isn’t CP or CPA a general abbreviation for cyclophosphamide?

Author Response

Point 1:

Figure 1 Please show abbreviation “M” and “G” in the Figure as “β-D-mannuronic acid (M)” and “α-L-guluronic acid (G)”.

Response 1:

Thank you for your comments, have revised

Point 2:

Line 152: The author describes “The TJ is the most apical linker complex, consisting of four unique transmembrane proteins: occludins, claudins, zonula occludens-1 (ZO-1), and junctional adhesion molecule (JAM)”, but ZO-1 is not transmembrane protein, and it exists in cytoplasm. Please correct that.

Response 2:

The sentence has been revised to “The TJ is the most apical linker complex, consisting of transmembrane proteins, such as occludin, claudins, and junctional adhesion molecule (JAM), and intracellular plaque proteins, such as zonula occludens (ZO)”. In the text, lines 202-205.

Point 3:

Line 212: Is Cy abbreviation for cyclophosphamide? Should describe non abbreviated word at the first appearance. Moreover, if Cy is abbreviation for cyclophosphamide, isn’t CP or CPA a general abbreviation for cyclophosphamide?

Response 3:

Cy is an abbreviation for cyclophosphamide, which is expressed differently in the abbreviations of different articles, and has been changed from the full text of cyclophosphamide to the general abbreviation CPA.

Reviewer 2 Report

This manuscript evaluates the effect of brown algae-derived polysaccharides in human health. The topic is interesting but in some points the manuscript is a bit limited. From my point of view, it should be interesting to add information about the main species of brown algae than that can be used for that purpose and where they can be found. It should be also interesting to add information the technological facility to obtain these polysaccharides from algae.

Line 19: This review.

Line 30: In the previous line the authors said that the intestine is an immune organ, so this comment is a redundancy. Delete the sentence until ;

Line 35: intestinal microbiota.

Line 41: Intestinal microbiota. The term flora in not actually used. Revise in the whole manuscript.

Line 43-45: This sentence is a repetition of the previous sentence. There is a lot of repetitions in Introduction section. The role in immune system and intestinal barrier is repeated several times. Please revise and be concise.

Line 51: Avoid the use of ;.

Line 54: All the genera in italics.

Table 1: Change source by Brown Algae. I think that the authors could include more studies in the table.

Figure 6: Please, briefly explain in the figure caption the protective mechanism described in the figure for a better interpretation. The l of Lactobacillus is in capital letter.

Author Response

Point 1:

It should be interesting to add information about the main species of brown algae than that can be used for that purpose and where they can be found. It should be also interesting to add information the technological facility to obtain these polysaccharides from algae.

Response 1:

Thanks to your comments, I have added information about the main species and distribution of brown algae with protective effects on the intestine and the methodological process for extracting the three brown algal polysaccharides in the text. They are in lines 64-88 and 152-163 of the text, respectively.

Point 2:

Line 19: This review.

Response 2:

Revised.

Point 3:

Line 30: In the previous line the authors said that the intestine is an immune organ, so this comment is a redundancy. Delete the sentence until ;

Line 35: intestinal microbiota.

Line 41: Intestinal microbiota. The term flora in not actually used. Revise in the whole manuscript.

Line 43-45: This sentence is a repetition of the previous sentence. There is a lot of repetitions in Introduction section. The role in immune system and intestinal barrier is repeated several times. Please revise and be concise.

Line 51: Avoid the use of ;.

Line 54: All the genera in italics.

Response 3:

Thank you for your comments, the introduction has been corrected and revised according to your comments.

Point 4:

Table 1: Change source by Brown Algae. I think that the authors could include more studies in the table.

Response 4:

More studies on the intestinal protective effects of brown algae polysaccharides have been added to Table 2 in line 184 of the text

Point 5:

Figure 6: Please, briefly explain in the figure caption the protective mechanism described in the figure for a better interpretation. The l of Lactobacillus is in capital letter.

Response 5:

The protective mechanisms in the figure are summarized and described in lines 438-457 of the text, and The l of Lactobacillus in the figure has been modified to a capital letter.

Round 2

Reviewer 2 Report

The authors have corrected the manuscript according my comments